# A Nutrigenetic Strategy for Reducing Blood Lipids and Low-Grade Inflammation in Adults with Obesity and Overweight

**DOI:** 10.3390/nu15204324

**Published:** 2023-10-10

**Authors:** Yolanda E. Pérez-Beltrán, Karina González-Becerra, Ingrid Rivera-Iñiguez, Erika Martínez-López, Omar Ramos-Lopez, Mildreth Alcaraz-Mejía, Roberto Rodríguez-Echevarría, Sonia G. Sáyago-Ayerdi, Edgar J. Mendivil

**Affiliations:** 1Laboratorio Integral de Investigación en Alimentos, Tecnológico Nacional de México/Instituto Tecnológico de Tepic, Tepic 63175, Nayarit, Mexico; yoelperezbe@ittepic.edu.mx; 2Departamento de Ciencias Médicas y de la Vida, Centro Universitario de la Ciénega, Instituto de Investigación en Genética Molecular, Universidad de Guadalajara, Ocotlán 47820, Jalisco, Mexico; karina.gbecerra@academicos.udg.mx; 3Department of Pediatrics, University of California, UCSD Center for Healthy Eating and Activity Research (CHEAR), San Diego, CA 92037, USA; iriverainiguez@health.ucsd.edu; 4Instituto de Nutrigenética y Nutrigenómica Traslacional, Departamento de Biología Molecular y Genómica, Centro Universitario de Ciencias de la Salud, Universidad de Guadalajara, Guadalajara 44340, Jalisco, Mexico; erika.martinez@academicos.udg.mx (E.M.-L.); roberto.rodriguez@academicos.udg.mx (R.R.-E.); 5Facultad de Medicina y Psicología, Universidad Autónoma de Baja California, Tijuana 22390, Baja California, Mexico; oscar.omar.ramos.lopez@uabc.edu.mx; 6Departamento de Electrónica, Sistemas e Informática, ITESO, Unioversidad Jesuita de Guadalajara, Tlaquepaque 45604, Jalisco, Mexico; mildreth@iteso.mx; 7Departamento de Salud, Universidad Iberoamericana, Ciudad de Mexico 01219, Mexico

**Keywords:** dyslipidemia, obesity, single-nucleotide polymorphism, nutrigenetic strategy, nutrigenetic portfolio, inflammatory markers

## Abstract

The pathogenesis of obesity and dyslipidemia involves genetic factors, such as polymorphisms related to lipid metabolism alterations predisposing their development. This study aimed to evaluate the effect of a nutrigenetic intervention on the blood lipid levels, body composition, and inflammation markers of adults with obesity and overweight. Eleven genetic variants associated with dyslipidemias in Mexicans were selected, and specific nutrigenetic recommendations for these polymorphisms were found. One hundred and one adults were recruited and assigned to follow either a standard or nutrigenetic diet for eight weeks. Anthropometric, biochemical, body composition, and inflammation markers were evaluated through standardized methods. Weighted genetic risk scores (wGRSs) were computed using the study polymorphisms. After intervention, both diets significantly decreased the anthropometric parameters and body composition (*p* < 0.05). Only the nutrigenetic diet group showed significant reductions in VLDL-c (*p* = 0.001), triglycerides (*p* = 0.002), TG:HDL (*p* = 0.002), IL-6 (*p* = 0.002), and TNF-α (*p* = 0.04). wGRSs had a high impact on the ΔTGs and ΔVLDL-c of both groups (standard diet: ΔTGs: Adj R^2^ = 0.69, *p* = 0.03; ΔVLDL-c: Adj R^2^ = 0.71, *p* = 0.02; nutrigenetic diet: ΔTGs: Adj R^2^ = 0.49, *p* = 0.03 and ΔVLDL-c: R^2^ = 0.29, *p* = 0.04). This nutrigenetic intervention improved lipid abnormalities in patients with excessive body weight. Hence, nutrigenetic strategies could be coadjuvant tools and enhance the standard dietary treatment for cardiometabolic diseases.

## 1. Introduction

Obesity and dyslipidemia are considered multifactorial and polygenic disorders with an alarming global proportion of prevalence [1]. In Mexico, overweight and obesity affect more than 75% of the adult population and more than 33% have a diagnosis of dyslipidemia [2,3]. Both pathologies represent risk factors for developing cardiovascular diseases and reducing the quality of life and life expectancy of an individual [4]. Standard guidelines for dyslipidemia and obesity management recommend to start with dietary strategies and physical activity and progress to pharmacological options [5]. Fortunately, several molecular and biochemical mechanisms and gene–diet interactions that trigger the metabolic alterations observed in both dyslipidemia and obesity have been elucidated in recent years.

Scientific evidence has shown that different feeding patterns have distinct molecular effects, including diverse impacts on the epigenome, leading to several physiological responses, i.e., diets can potentially compensate or accentuate disease states according to an individual’s genetic background [6,7,8]. It has been recognized that individuals have distinct metabolic responses depending on their genotype due to the presence or absence of genetic variants that confer protection against disease, adaptive advantage, or disease susceptibility [9]. Thus, genetic variations in specific nucleotides (single-nucleotide polymorphisms (SNPs)) associated with obesity and dyslipidemia are crucial factors behind the predisposition to suffer from cardiometabolic diseases [10]. Up to now, different SNPs related to alterations in lipid metabolism have been identified [11]. Recently, a panel of lipid-related genetic variants for personalized dietary interventions was proposed by this group [12]. Some SNPs of the panel (*LIPC* rs1800588, *LPL* rs13702, *FABP2* rs1799883, *ABCA1* rs9282541, *CETP* rs708272, *PPARG* rs1801282, *APOA5* rs662799, *APOC3* rs5128, *APOA1* rs670, *APOE* rs7412, and rs429358) have a high prevalence in the Mexican population and are associated with the most common dyslipidemias in this population [13,14,15]. Also, specific genetic risk scores (GRSs), including multiple genetic loci, have explained some populations’ susceptibility to metabolic risks [16]. However, it is essential to consider that the prevalence of genetic variants differs by ancestry; for that reason, efforts to study different ancestries are worthwhile.

Parallel to these findings, precision nutrition has gained prominence, and nutrigenetic recommendations have been identified as potential tools that complement the standard dietary recommendations for the prevention and precision management of weight loss, dyslipidemia, and many more metabolic disorders or pathologies [17,18]. The nutrigenetic recommendations contemplate the precise nutritional requirements of an individual, considering their genetic characteristics, which are present throughout the life cycle and can be inherited by offspring [19]. Nutrigenetic recommendations could be classified and grouped in nutrigenetic portfolios. These sets of dietetic recommendations are based on scientific evidence that represents a basis for creating nutrigenetic patterns established by the genetic and physiological characteristics of different populations worldwide. The application of precision nutrition holds great promise for providing better nutritional advice to the general public, genetic subgroups, and individuals, in contrast to following conventional dietary recommendations, whose effect may be limited. Hence, it is crucial to identify and implement nutrigenetic recommendations that support the clinical practice of nutrition specialists to improve the population’s health. Therefore, this research aimed to evaluate the effects of a nutrigenetic clinical intervention on the cardiometabolic markers and body composition in adults with overweight and obesity.

## 2. Materials and Methods

### 2.1. Study Population

This controlled clinical trial included 101 subjects with overweight or obesity, with a BMI 25 ≥ 40 (kg/m^2^), who were from 18 to 50 years old, with a waist circumference >94 cm for men and >80 cm for women, with Mexican ancestry, and who were residents of the metropolitan area of Guadalajara, Jalisco, Mexico, who voluntarily decided to participate in the nutrigenetic intervention after finding advertising flyers on social networks. The exclusion criteria were as follows: a history of cardiovascular disease; a diagnosed mental illness or intellectual disability; pregnant or lactating women; subjects with gastrointestinal disorders (pancreatitis, colitis, irritable bowel syndrome, Crohn’s disease); diagnosed with diabetes, endocrine disorders, or cancer; and individuals with a fasting glucose >100 mg/dL that were not taking lipid- or weight-lowering medication, sulfonylureas, thiazolidinediones, insulin, glucocorticoids, antineoplastic agents, angiotensin receptor blockers, angiotensin-converting enzyme inhibitors, or psychoactive medication. Volunteers were recruited and evaluated at the Institute of Nutrigenetics and Translational Nutrigenomics (INNUGET), University of Guadalajara, and at the Department of Psychology, Education and Health, ITESO, Jesuit University of Guadalajara.

All subjects signed an informed consent form before participating in this study, which was conducted according to the principles expressed in the Declaration of Helsinki. The trial procedures were approved by the ethics committee of ITESO, Jesuit University of Guadalajara, and registered under number 0001DRC. (ClinicalTrials.gov: NCT05210023).

### 2.2. Genotyping

Before the intervention, a peripheral blood sample was taken from enrolled participants for genotyping at the INNUGET facilities. According to the manufacturer’s instructions, DNA was extracted from leucocytes using a DNA extraction kit (High Pure PCR Template Preparation kit, Roche Diagnostics, Mannheim, Germany). The concentration and purity of the genomic DNA were assessed using a microvolume spectrophotometer (Multiskan SkyHigh, Thermofisher Scientific, Waltham, MA, USA). The genotypes of interest (*LIPC* rs1800588, *LPL* rs13702, *FABP2* rs1799883, *ABCA1* rs9282541, *CETP* rs708272, *PPARG* rs1801282, *APOA5* rs662799, *APOC3* rs5128, *APOA1* rs670, *APOE* rs7412, and rs429358) were detected using a real-time PCR system using allelic discrimination in a 96-well format (Appendix A) and were read using a Roche LightCycler 96 system (Roche, Mannheim, Germany). The PCR was carried out with an initial denaturation time of 10 min at 95 °C followed by an annealing/extension at 60 °C for 60 s (40 cycles).

### 2.3. Nutrigenetic Portfolios

A nutrigenetic portfolio was structured according to a compendium of weekly menus designed on general dietary recommendations proposed for the treatment of obesity and dyslipidemia [20,21,22] or based on lipid-related nutrigenetic recommendations previously identified by this research group [17]. Thus, the set of menus was classified into two groups: (1) standard diet menus (50% carbohydrates, 25% protein, and 25% fat: 6–11% polyunsaturated fatty acids (PUFAs), 15% monounsaturated fatty acids (MUFAs), and >10% saturated fatty acids (SAT)); (2) nutrigenetic portfolios: Pattern 1 (50% low glycemic index carbohydrates, 25% protein, and 25% fat: 10% PUFAs, 10% MUFAs, and 5% SAT) and Pattern 2 (45% carbohydrates, 20% protein, and 35% fat: 15% PUFAs, 15% MUFAs, and 5% SAT).

From each menu classification, four different weekly options were structured and standardized (ESHA’s Food Processor^®^ Nutrition Analysis software, version 11.11, Rockford, IL, USA), ranging from 1000 to 2200 kcal, and included five mealtimes (breakfast, morning snack, lunch, evening snack, and dinner).

Every 15 days, the participants received a personalized nutritional consultation online, and a new menu was provided.

### 2.4. Dietary Intake and Assessment

A stratified randomization was carried out considering the sex, age, and BMI of the enrolled participants, and they were divided into two study groups to follow either a standard diet or a nutrigenetic diet for eight weeks. The energy prescription for the individuals of both study groups was calculated according to 25–30 kcal/kg of the ideal weight suggested for patients with overweight and obesity, with a caloric restriction of ~500 kcal [20]. The menus provided to the participants were obtained from the nutrigenetic portfolio, according to the corresponding treatment. In the case of the nutrigenetic diet group, a predictive mathematical algorithm was used to identify which nutrigenetic pattern was most suitable for each individual based on their genotypes. We assigned a ponderation to each gene to find the correct prediction equation, considering the available scientific evidence of the gene and SNP, the allele frequency in the Mexican population, and the relevance of the gene with dyslipidemia in this population. The gene ponderation, combined with the variable of each corresponding genotype, could lead in a negative, positive, or neutral way to the decision of the corresponding pattern.

A 3-day diet record was used to assess the daily intake of macronutrients and fatty acids. Each subject was instructed on how to complete this record, including two weekdays and one weekend day. The food records were coded by trained dietitians using ESHA’s Food Processor^®^ Nutrition Analysis software (version 11.11, Rockford, IL, USA). Nutrient intakes were averaged over the 3-day food records and adherence to the diet was evaluated through the percentage of adequacy: %AD = [consumed daily intake (g)/recommended daily intake (g)] × 100.

### 2.5. Anthropometric and Body Composition Measurements

Anthropometric measurements, including body weights (kg), heights (cm), and waist and hip circumferences (WC and HC; cm), were collected by trained dietitians following conventional validated methods [23]. Total body fat (TBF; % and kg), visceral fat index (VFAT), skeletal muscle mass (MuscleM; g), and body mineral mass (MinM; g) contents were determined using electrical bioimpedance (InBody 120, Segmental Multi-frequency Body DSM-BIA, Seoul, Republic of Korea). Systolic and diastolic blood pressures were quantified with an automated sphygmomanometer according to the standardized criteria of the International Society of Hypertension [24]. Anthropometric parameters were evaluated three times throughout the intervention: at the baseline, midterm point (week 4), and final point (week 8).

The participants were instructed not to perform physical activity and remained sedentary during the intervention.

### 2.6. Biochemical Measurements

Peripheral blood samples were taken after a 12 h overnight fast and immediately centrifuged at 3500 rpm to obtain serum. Biochemical variables, the glucose (mg/dL), total cholesterol (TC, mg/dL), triglycerides (TGs, mg/dL), high-density lipoprotein cholesterol (HDL-c, mg/dL), very low density lipoprotein cholesterol (VLDL-c, mg/dL), and C-reactive protein (CRP, mg/L), were analyzed using a dry chemistry analyzer (Vitros 250 Analyzer, Ortho-Clinical Diagnostics, Johnson & Johnson Services, Inc., Rochester, NY, USA). Low-density lipoprotein cholesterol (LDL-c, mg/dL) was calculated using the Friedewald formula [25]. Serum insulin levels (μUI/mL) were determined using a LIASON^®^ immunoassay kit, (Insulin ref: 310360 Diasorin Liaison^®^, Milan, Italy). Homeostatic model assessment of insulin resistance (HOMA-IR) was calculated from fasting blood plasma glucose and serum insulin values [26]. An HOMA-IR value >2.5 was considered insulin resistance. The TGL:HDL ratio was calculated with the following formula: fasting TGL mg/dL/fasting HDL-c mg/dL; a ratio >3 was considered insulin-resistant [27]. The parameters mentioned above were evaluated on three occasions. 

### 2.7. Inflammation Markers

Interleukin 6 (IL-6, pg/mL) and TNF-α (pg/mL) concentrations were determined using ProQuantum High Sensitivity Immunoassay kits (Ref: A35601, A35573 Invitrogen by Thermo Fisher Scientific, Carlsbad, CA, USA) at the baseline and at the end of the intervention (week 8).

### 2.8. Statistical Analyses

Data are expressed as mean ± standard error. The normality of the variables was analyzed using the Kolmogorov–Smirnov test and the equality of the variances with the Levene test.

Paired and unpaired Student’s *t*-tests were applied. ANOVA repeated measures with post hoc analysis (Tukey’s test) were used to evaluate the difference between evaluation times. 

The genetic variation in the population was tested for Hardy–Weinberg equilibrium (HWE) using a χ^2^ test. Data analysis was conducted using the statistical package IMB SPSS version 26 (Statistical Package for the Social Sciences, Chicago, IL, USA); *p* < 0.05 was considered statistically significant.

The sample size was calculated considering that the primary outcome measure was the change in the fasting plasma total cholesterol. Assuming a 20% post-diet difference and 0.99 SD, with α = 0.05 and 1 − β = 0.8, a minimum of 56 participants (28 per treatment) were required and 30% plus were considered for desertions. The aim was to recruit a total of 100 subjects.

### 2.9. Genetic Risk Score (GRS) Calculation

Multiple linear regression models were used to predict changes in the primary outcomes obtained in each dietary group, considering the independent effect of each SNP under study. For this purpose, genotypes of each SNP were differentially coded into two categories: “non-risk = 0” (absence of risk allele) and “risk = 1” (presence of risk allele). A Bonferroni test was applied for multiple comparison correction. 

Student’s *t*-tests were run to the categorized genotype groups (risk vs. non-risk) of each dietary treatment to assess statistical differences (*p* < 0.05). In addition, squared partial correlations (PC^2^’s) were performed to estimate each SNP’s individual contribution to the prominent outcome variability in response to diets.

In addition, to evaluate the additive effects of each SNP on the main outcomes for dietary groups (standard and nutrigenetic diet), weighted genetic risk scores (wGRSs) were constructed through additive models [28].The wGRS was computed by multiplying the number of high-risk genotypes at each locus for the corresponding effect sizes (β-coefficients) and then summing the products. The derived scores were based on the assumption that the included SNPs contribute additively to explain the main outcomes and have independent effects. 

Statistical analyses were performed in the statistical program STATA 12 (StataCorp LLC., College Station, TX, USA; www.stata.com). The statistical significance was set at *p* < 0.05.

## 3. Results

### 3.1. Demographic, Anthropometric, and Body Composition Characteristics of the Study Population

The trial was conducted from August 2021 to December 2021 in Guadalajara, Jalisco, Mexico, at the Institute of Nutrigenetics and Translational Nutrigenomics, University of Guadalajara, and at the Department of Psychology, Education and Health, ITESO, Jesuit University of Guadalajara. One hundred and one individuals were randomly assigned to the following treatments: a standard diet (n = 51) and a nutrigenetic diet (n = 50 individuals). Figure 1 shows the flow diagram of the enrollment and follow-up of the study participants. Fifteen participants from the standard diet group and nine from the nutrigenetic diet dropped out of the intervention. Twenty-three individuals (23%) abandoned the intervention, mainly because they got sick from COVID-19 or had to care for an infected family member. Thus, 78 participants finished the intervention. 

Table 1 shows the study subjects’ demographic, anthropometric, and body composition characteristics. The mean age of the total population was 33.5 ± 9.0 years; 51% of the population were women (n = 52), and 49% were men (n = 49). No significant differences between groups were found regarding the baseline demographic, anthropometric, and body composition variables.

At the baseline, parameters, such as the BMI (31.46 ± 0.7 and 29.86 ± 0.5 kg/m^2^), VFAT (15.22 ± 0.6 and 14.55 ± 0.6), WC (96.88 ± 2.2 and 93.31 ± 1.7 cm), and HC (110.84 ± 1.5 and 109.83 ± 1.7 cm) of both groups (standard diet and nutrigenetic diet, respectively), were above the reference values established as healthy (BMI: 18.5–24.9; VFAT: <9; HC men: <94 cm, women <80 cm).

Throughout the intervention, the body weight, BMI, total body fat mass (kg and %), VFAT, WC, and HC, showed significant favorable changes (*p* < 0.05) regardless of the type of diet followed (Table 1). 

Body weight, BMI, and total body fat mass (kg and %) reductions in the nutrigenetic diet group were present in the three study time points, with significant differences between them. In contrast, only significant changes were detected between the baseline and the 4-week measurements in the standard diet group. No statistical differences were observed between deltas Δ (baseline–final value) for anthropometric parameters; however, the delta (Δ) of body weight (kg) and body fat (kg) of the nutrigenetic diet group showed higher values than the standard diet group, which represents a greater clinical significance (nutrigenetic diet = −4.25 ± 0.5 kg and −3.09 ± 0.4 kg; standard diet = −3.02 ± 0.5 kg and −2.55 ± 0.3 kg, Δ of body weight and body fat, respectively).

The standard diet group participants’ skeletal muscle mass (kg) remained stable during the intervention. The skeletal muscle mass was significantly reduced (Bsl vs. 4-week −0.50 ± 0.1 kg) in the nutrigenetic group (Table 1). Nevertheless, this parameter was stabilized in the final evaluation; the skeletal muscle mass loss was not sustained. 

The body mineral mass (kg) of participants in both groups did not change throughout the intervention.

### 3.2. Biochemical Profile and Inflammation Markers

Table 2 shows the changes in the biochemical parameters of participants. At the baseline, only HDL-c, VLDL-c, triglycerides, C-reactive protein, and HOMA-IR were out of the normal range. The nutrigenetic diet group showed more significant reductions in VLDL-c (standard diet: *p* = 1.199 vs. nutrigenetic diet: *p* ≤ 0.001), triglycerides (*p* = 0.209 vs. *p* = 0.002), and the HDL:TGL ratio (*p* = 0.275 vs. *p* = 0.002).

No significant difference in HDL-c levels was detected between groups (standard diet: *p* = 0.922 and nutrigenetic diet: *p* = 0.062), but a positive tendency to increase HDL-c was observed in the group that followed the nutrigenetic diet. The Δ result showed a gain of +1.5 mg/dL in the nutrigenetic diet vs. the Δ of the standard diet group, which was −0.08 mg/dL.

The nutrigenetic diet group showed significant changes when comparing baseline vs. final TG values (*p* = 0.002). The decrease of −32.69 ± 9.6 mg/dL (20%) is considered of great clinical relevance since the therapy target proposed by the guidelines on managing blood cholesterol was met. No significant changes in TG concentrations were observed in the standard diet group.

Participants in both groups had slightly altered VLDL-c baseline values. No significant changes were reported in the standard diet group. But a significant decrease in VLDL-c was observed in the nutrigenetic diet group (Δ = −6.6 ± 1.9 mg/dL = −20%). 

A statistical difference was detected in the TGL:HDL ratio between groups; while this parameter remained unchanged in the standard diet group, there was a significant reduction in the nutrigenetic diet group (Δ = −1.09 ± 0.4, *p* = 0.002).

The baseline glucose values of the participants in both groups were normal, and the baseline insulin values were slightly above the healthy range. Throughout the intervention, a significant improvement was observed in glucose (Δ standard diet −3.55 ± 1.5, *p* = 0.02; Δ nutrigenetic diet: −3.73 ± 1.4, *p* ≤ 0.01) and insulin concentrations (Δ standard diet −4.55 ± 1.7, *p* = 0.01; Δ nutrigenetic diet: −4.66 ± 1.4, *p* ≤ 0.002) in the participants of both groups, and, consequently, the HOMA-IR decreased significantly.

No changes in C-reactive protein concentration were observed throughout the intervention. The baseline values of IL-6 (1.74 ± 0.4 pg/mL; 1.69 ± 0.2 pg/mL) and TNF-α (0.55 ± 0.1 pg/mL; 0.84 ± 0.2) of the standard and nutrigenetic diets, respectively, were within the parameters considered healthy (IL-6 < 3.2 pg/mL and TNF-α < 3.7 pg/mL); even the TNF-α concentrations of some individuals in both groups were below the limit of quantification. In the participants who followed the nutrigenetic diet, there was a significant decrease in circulating IL-6 when comparing baseline vs. final concentrations (1.69 ± 0.29 to 0.89 ± 0.18 pg/mL, *p* = 0.002, Δ = −0.80 ± 0.22 and TNF- α (0.84 ± 0.25 to 0.47 ± 0.26, *p* = 0.043, Δ = −0.37 ± 0.10), while no significant changes were observed in the standard diet group. No statistical difference between treatments was detected.

**Table 1 nutrients-15-04324-t001:** Demographic, anthropometric, and body composition characteristics of study subjects.

**Parameter**	**Total Population (n = 101)**		**Standard Diet (n = 36)**			**Nutrigenetic Diet (n = 42)**				***p* Value**
Age (years)	33.5 ± 9.0			32.8 ± 8.5			34.1 ± 9.5			NS
Men	49 (49%)			26 (51%)			23 (46%)			NS
Women	52 (51%)			25 (49%)			27 (54%)			NS
	Standard diet (n = 36)					Nutrigenetic Diet (n = 42)				
**Anthropometrics and Body Composition**										
**Parameter**	**Baseline**	**4 Weeks**	**8 Weeks**	**Δ**	***p* Value**	**Baseline**	**4 Weeks**	**8 Weeks**	**Δ**	***p* Value**
Body weight (kg)	88.0 ± 2.5 ^a^	85.65 ± 2.4 ^b^	84.98 ± 2.4 ^b^	−3.02 ± 0.5	<0.001	84.92 ± 2.3 ^a^	82.24 ± 2.2 ^b^	81.04 ± 2.2 ^c^	−4.25 ± 0.5	<0.001
BMI (kg/m^2^)	31.46 ± 0.7 ^a^	30.64 ± 0.7 ^b^	30.38 ± 0.7 ^b^	−1.37 ± 0.5	<0.001	29.86 ± 0.5 ^a^	28.87 ± 0.5 ^b^	28.44 ± 0.5 ^c^	−1.41 ± 0.1	<0.001
Body fat (kg)	33.91 ± 1.5 ^a^	32.24 ± 1.5 ^b^	31.35 ± 1.5 ^b^	−2.55 ± 0.3	<0.001	31.30 ± 1.2 ^a^	29.42 ± 1.2 ^b^	28.20 ± 1.2 ^c^	−3.09 ± 0.4	<0.001
Body fat (%)	38.27 ± 1.1 ^a^	37.41 ± 1.2 ^ab^	36.65 ± 1.2 ^b^	−1.62 ± 0.3	<0.001	36.66 ± 1.0 ^a^	35.49 ± 1.0 ^b^	34.36 ± 1.1 ^c^	−2.30 ± 0.4	<0.001
VFAT	15.22 ± 0.6 ^a^	14.56 ± 0.6 ^b^	14.08 ± 0.7 ^b^	−1.13 ± 0.2	<0.001	14.55 ± 0.6 ^a^	13.67 ± 0.6 ^b^	13.12 ± 0.6 ^c^	−1.42 ± 0.2	<0.001
WC (cm)	96.88 ± 2.2 ^a^	93.51 ± 1.9 ^b^	92.95 ± 2.1 ^b^	−3.93 ± 0.7	<0.001	93.31 ± 1.7 ^a^	90.43 ± 1.4 ^b^	89.05 ± 1.6 ^b^	−4.25 ± 0.7	<0.001
HC (cm)	110.84 ± 1.5 ^a^	108.88 ± 1.4 ^b^	107.74 ± 1.4 ^b^	−3.10 ± 0.4	<0.001	109.83 ± 1.7 ^a^	108.11 ± 1.0 ^b^	106.94 ± 1.1 ^b^	−2.88 ± 0.4	<0.001
SMM (kg)	30.32 ± 1.0 ^a^	30.00 ± 0.9 ^a^	30.02 ± 1.0 ^a^	−0.29 ± 0.1	0.067	30.01 ± 1.0 ^a^	29.61 ± 1.0 ^b^	29.51 ± 1.0 ^b^	0.40 ± 0.1	0.057
BMM (kg)	3.75 ± 0.1 ^a^	3.71 ± 0.1 ^a^	3.72 ± 0.1 ^a^	−0.03 ± 0.1	0.085	3.78 ± 0.1 ^a^	3.70 ± 0.1 ^a^	3.71 ± 0.1 ^a^	−0.06 ± 0.1	0.074

Age (years) is presented as mean ± SD; values are presented as mean ± SEM or percentage; NS: no significant difference; Δ = final value—basal; BMI: body mass index; VFAT: visceral fat index; WC: waist circumference; HC: hip circumference; SMM: skeletal muscle mass; BMM: body mineral mass. Comparisons between groups (baseline vs. final) were made with a paired student’s *t*-test. Comparisons between groups (standard diet vs. nutrigenetic diet) were made with a student’s *t*-test for independent samples (no statistically significant differences between groups were detected). ^a.b.c^: significant differences between evaluation times (*p* < 0.05). Post hoc differences between the means were detected using the Tukey HSD test.

**Table 2 nutrients-15-04324-t002:** Changes in biochemical profile and inflammation markers of the population throughout the intervention.

	Standard Diet (n = 36)					Nutrigenetic Diet (n = 42)				
Parameter	Baseline	4 Weeks	8 Weeks	Δ	*p* Value	Baseline	4 Weeks	8 Weeks	Δ	*p* Value
TC (mg/dL)	172.80 ± 5.5 ^a^	154.97 ± 5.8 ^b^	158.50 ± 5.3 ^b^	−14.3 ± 3.5	**<0.001**	177.90 ± 4.4 ^a^	156.40 ± 4.7 ^b^	161.31 ± 4.1 ^b^	−16.59 ± 3.0	**<0.001**
LDL-c (mg/dL)	104.58 ± 4.6 ^a^	93.05 ± 4.6 ^b^	94.00 ± 4.1 ^b^	−10.58 ± 3.0	**<0.001**	108.31 ± 3.9 ^a^	98.16 ± 3.8 ^b^	99.31 ± 3.59 ^b^	−9.00 ± 3.1	**0.006**
HDL-c (mg/dL)	35.02 ± 1.6 ^a^	33.33 ± 1.5 ^a^	34.94 ± 1.6 ^a^	−0.08 ± 0.8	0.922	35.0 ± 1.3 ^a^	35.5 ± 1.0 ^a^	36.5 ± 1.2 ^a^	+1.5 ± 0.8	0.062
TG (mg/dL) *	166.02 ± 14.1 ^a^	142.47 ± 14.6 ^a^	152.25 ± 13.6 ^a^	−13.7 ± 10.7	0.209	160.69 ± 11.3 ^a^	129.66 ± 8.9 ^b^	128.00 ± 8.3 ^b^	−32.69 ± 9.6	**0.002 ***
VLDL- (mg/dL) *	33.19 ± 2.8 ^a^	28.50 ± 2.9 ^a^	30.38 ± 2.7 ^a^	−2.8 ± 2.1	0.199	32.21 ± 2.8 ^a^	25.90 ± 1.8 ^b^	25.61 ± 1.6 ^b^	−6.6 ± 1.9	**<0.001 ***
TG:HDL *	4.74 ± 0.6 ^a^	4.27 ± 0.7 ^a^	4.35 ± 0.6 ^a^	−0.39 ± 0.3	0.275	4.59 ± 0.4 ^a^	3.65 ± 0.3 ^b^	3.50 ± 0.3 ^b^	−1.09 ± 0.4	**0.002 ***
Glucose (mg/dL)	90.08 ± 2.1 ^a^	85.52 ± 2.1 ^b^	86.52 ± 2.5 ^ab^	−3.55 ± 1.5	**0.025**	89.11 ± 1.6 ^a^	82.28 ± 1.3 ^b^	85.38 ± 1.4 ^ab^	−3.73 ± 1.4	**<0.012**
Insulin (μUI/mL)	19.25 ± 1.8 ^a^	15.00 ± 1.6 ^ab^	14.69 ± 1.0 ^b^	−4.55 ± 1.7	**0.014**	18.33 ± 2.1 ^a^	14.04 ± 1.2 ^b^	13.66 ± 1.7 ^b^	−4.66 ± 1.4	**<0.002**
HOMA-IR	4.61 ± 0.4 ^a^	3.54 ± 0.4 ^ab^	3.43 ± 0.3 ^b^	−1.18 ± 0.4	**0.032**	4.12 ± 0.5 ^a^	2.89 ± 0.2 ^b^	2.94 ± 0.4 ^b^	−1.18 ± 0.3	**0.029**
CRP (mg/L)	11.00 ± 5.0 ^a^	10.36 ± 4.6 ^a^	10.08 ± 3.4 ^a^	−0.91 ± 4.9	0.276	11.21 ± 7.8 ^a^	10.97 ± 7.06 ^a^	11.16 ± 8.4 ^a^	−0.04 ± 8.9	0.373
IL-6 (pg/mL) (n = 20)	1.74 ± 0.45 ^a^		1.30 ± 0.34 ^a^	−0.44 ± 0.20	0.86	1.69 ± 0.29 ^a^		0.89 ± 0.18 ^b^	−0.80 ± 0.22	**0.002**
TNF-α (pg/mL) (n = 20)	0.55 ± 0.10 ^a^		0.34 ± 0.09 ^a^	−0.21 ± 0.12	0.109	0.84 ± 0.25 ^a^		0.47 ± 0.26 ^b^	−0.37 ± 0.10	**0.043**

Data presented as mean ± SEM; bold numbers indicate *p* < 0.05; Δ = final value–−basal. TC: total cholesterol; LDL-c: low-density lipoprotein cholesterol; HDL-c: high-density lipoprotein cholesterol; TGs: triglycerides; VLDL-c: very low density lipoprotein cholesterol; TG:HDL: triglycerides: high-density lipoprotein cholesterol index; HOMA-IR: homeostasis model assessment; CRP: C-reactive protein; IL-6: interleukin 6; TNF-α: tumor necrosis factor alpha. Comparisons between groups (baseline vs. final) were made with a paired student’s *t*-test. Comparisons between groups (standard diet vs. nutrigenetic diet) were made with a student’s *t*-test for independent samples (***** statistically significant differences between groups). ^a.b^: significant differences between evaluation times (*p* < 0.05). Post hoc differences between the means were detected using the Tukey HSD.

### 3.3. Dietary Compliance

Table 3 summarizes the compliance to treatment, showing the kcal; macronutrients; and saturated, monounsaturated, and polyunsaturated fats in both treatments. The overall adherence to both diets was good (91.2 ± 2.3–114.4 ± 3.9%), with no difference between the standard and the nutrigenetic groups regarding macronutrients. However, the nutrigenetic diet group had an overconsumption of saturated fats (181.0 ± 13.8%, *p* = 0.005) and an underconsumption of MUFAs (64.1 ± 6.2%, *p* = 0.009) by the standard diet participants. Therefore, these two variables (SatFats and MUFAs) were considered adjustment variables.

Regarding the PUFA intake, both groups consumed a lower amount relative to the recommended daily intake (standard diet: 81.5 ± 10.8% and nutrigenetic diet: 66.1 ± 6.0%), with no significant difference between treatments (*p* = 0.19)

**Table 3 nutrients-15-04324-t003:** Assessment of dietary compliance (%).

Component	Standard Diet (n = 22)	Nutrigenetic Diet (n = 35)	*p* Value
Energy	97.0 ± 2.1	98.0 ± 5.8	0.89
Protein	114.4 ± 3.9	96.7 ± 6.1	0.09
Carbohydrate	91.2 ± 2.3	95.5 ± 6.9	0.61
Fat	100.3 ± 4.0	113.9 ± 7.9	0.18
Saturated fat	125.1 ± 9.2	181.0 ± 13.8	**0.005**
Monounsaturated fat	64.1 ± 6.2	91.8 ± 7.1	**0.009**
Polyunsaturated fat	81.5 ± 10.8	66.1 ± 6.0	0.19

Consumption (%) of the prescribed daily intake of dietary components. Values are mean ± SEM. Bold numbers indicate *p* < 0.05. *p* value: difference between dietary groups.

### 3.4. Genetic Risk Score (GRS)

The distribution of the genotypes under study was concordant with the Hardy–Weinberg equilibrium principle (Appendix A).

The multiple linear regression models considered the SNPs that explained ΔTGs and ΔVLDL-c outcomes by dietary groups, and they are presented in Table 4 and Table 5, correspondingly. Three SNPs, specific for each dietary intervention, were significantly associated with TG and VLDL-c outcomes (standard diet: *LPL* rs13702 and *APOA1* rs670; nutrigenetic diet: *LIPC* rs1800588). The predictive model for standard diet consumption showed that LPL rs13702 has a deleterious effect on ΔTGs (β = 159.13 mg/dL, *p* = 0.013) as well as on ΔVLDL (β = 31.35 mg/dL, *p* = 0.012), whereas *APOA1* rs670 showed a beneficial effect on ΔTGs (β = −112.95 mg/dL, *p* = 0.013 and ΔVLDL-c (β = −22.39 mg/dL, *p* = 0.012). 

The individual contributions of each predictor to the models are also reported (Table 4 and Table 5). Interestingly, *LPL* rs13702 showed 26% of the TG and VLDL-c variance (TGs: PC^2^ = 0.26, *p* = 0.01; VLDL-c: PC^2^ = 0.26, *p* = 0.01) in the standard diet group, while *APOA1* rs670 predicted a 25–26% reduction in TGs and VLDL-c (PC^2^ = 0.25; PC^2^ = 0.26, *p* = 0.01) when consuming the standard diet.

No common SNPs related to TG and VLDL-c variabilities between diets were found. In this sense, predictive models for a nutrigenetic diet (Table 5) showed that *LIPC* rs1800588 influenced the variability in TGs (β = 34.38 mg/dL, *p* = 0.028) and VLDL-c (β = 6.82 mg/dL, *p* = 0.029). *LIPC* rs1800588 predicted 14% and 15% of the TG and VLDL-c variance in the nutrigenetic diet (PC^2^ = 0.14, *p* = 0.02; PC^2^ = 0.15, *p* = 0.02).

The predictive performance of the wGRS on TGs and VLDL-c is also reported in Table 4 and Table 5. It was observed that wGRSs were the most significant contributor to the variability in TG and VLDL-c levels. In the standard diet group, the wGRS showed an impressive effect of 69% (Adj R^2^ = 0.69, *p* = 0.03) on ΔTGs and 71% on ΔVLDL-c (Adj R^2^ = 0.71, *p* = 0.02). Concerning the predictive effect of wGRSs on ΔTGs and ΔVLDL-c of the nutrigenetic diet group, impacts of 49% (Adj R^2^ = 0.49, *p* = 0.03) and 29% (Adj R^2^ = 0.29, *p* = 0.04) were reported, respectively.

To avoid collinearity, we chose to construct the wGRS using only SNPs from genes that were not highly correlated (*ABCA1* rs9282541, *APOA5* rs662799, and *APOC3* rs5128 were omitted for collinearity).

**Table 5 nutrients-15-04324-t005:** Multiple linear regression models using the panel of SNPs to predict the Δ of TGs and VLDL when consuming a nutrigenetic diet.

Nutrigenetic Diet										
	Model 1					Model 2				
	ΔTGs					ΔVLDL				
Predictors	β Coefficients	Std. Error	*p* Value ^a^	PC^2^	*p* Value ^b^	β Coefficients	Std. Error	*p* Value ^a^	PC^2^	*p* Value ^b^
*LIPC* rs1800588	34.38	14.45	**0.028**	0.14	**0.02**	6.82	2.88	**0.029**	0.15	**0.02**
*LPL* rs13702	10.68	24.32	0.665	0.005	0.66	2.08	4.85	0.672	0.005	0.67
*FABP2* rs1799883	−21.62	20.44	0.294	0.03	0.29	−3.90	4.00	0.341	0.02	0.34
*CETP* rs708272	−47.55	24.63	0.069	0.09	0.06	−9.28	4.91	0.074	0.09	0.07
*APOE* rs7412 and rs429358	−28.74	17.16	0.110	0.07	0.11	−5.36	3.42	0.134	0.06	0.13
SATFAT	0.98	2.23	0.663	0.005	0.66	0.18	0.44	0.680	0.004	0.68
MUFAs	−1.72	2.08	0.420	0.01	0.41	−0.33	0.41	0.432	0.017	0.43
Adj. R^2^	0.49		**0.03**			0.29		**0.04**		

Data are expressed as β values ± standard error. Bold numbers indicate *p* < 0.05. PC^2^: squared partial correlation; ΔTGs: delta of triglycerides; ΔVLDL: delta of very low density lipoprotein; *p* value ^a^ = *p* of each predictor; *p* value ^b^ = *p* of PC^2^. Omitted for collinearity: *ABCA1* rs9282541, *APOA5* rs662799, *APOC3* rs5128, *PPARG* rs1801282, and *APOA1* rs670.

Carriers of the risk allele of *LIPC* rs1800588 (CT + TT) present a greater reduction in TGs and VLDL-c than non-carriers (CC) consuming a nutrigenetic diet (ΔTGs 73.42 ± 26.72, *p* = 0.006; ΔVLDL 14.75 ± 5.32, *p* = 0.006). That is, the carriers of the risk allele of *LIPC rs1800588* showed greater statistical decreases in TGs and VLDL-c after ingesting the nutrigenetic diet. However, no significant difference was observed in the reduction in TGs and VLDL-c of the carriers and non-carriers of the risk alleles of *LPL* rs13702 (TC + CC vs. TT) and *APOA*1 rs670 (GA + AA vs. GG) when consuming a standard diet (Appendix A).

## 4. Discussion

Abnormal anthropometric and body composition markers are associated with CVD and other metabolic alterations [29]. Scientific evidence indicates that different biological factors can influence body weight, body composition, and biochemical parameters between individuals [30]. Among these factors, some SNPs have been associated with changes in lipid metabolism in response to diverse dietary prescriptions [17].

Therefore, in the present study, we investigated the effect of two functional diet treatments, a standard diet and a nutrigenetic diet, on anthropometric parameters, body composition, biochemical parameters, and inflammation markers of Mexican adults with overweight and obesity considering a panel of 10 genes and 11 polymorphisms associated with dyslipidemia in Mexicans. 

Participants were randomized to either a standard diet (with gold standard recommendations for obesity and dyslipidemia management) [20] or to a nutrigenetic diet (with recommendations according to their genetic background published elsewhere) [17]. Participants significantly reduced their anthropometric parameters regardless of treatment. However, the body weight, BMI, total body fat mass (kg and %), and visceral fat index significantly decreased across the three study time points in the nutrigenetic group. In contrast, significant differences in the standard diet group were only present between the baseline and the intermediate measurement. Moreover, there was a prominent clinical reduction in body fat (kg, %) and VFAT, both risk factors that trigger multiple chronic non-communicable diseases. These results show a more significant effect of the nutrigenetic diet compared with the standard diet group, regarding anthropometric and body composition improvements.

Similarly, other nutrigenetic interventions have shown changes in anthropometric parameters when evaluated more than two times [31,32]. Moreover, Martinez-Lopez et al. and de Luis et al. [31,33] suggest that the polymorphisms of the participant influence the effect of the diet, so if a precision diet is provided, metabolic changes will be more favorable. It is essential to mention that the caloric deficit to which the participants were subjected led to the reductions in their anthropometric parameters since, when the caloric intake is less than caloric expenditure, body weight loss occurs [6].

Regarding muscle mass, scientific evidence indicates that exposure to an uncontrolled caloric deficit commonly decreases muscle protein synthesis in humans since at least a part of the energy and the substrates usually assigned to protein synthesis muscles is diverted elsewhere when energy is low [34]. Nevertheless, this phenomenon was not observed in the participants of the standard diet, but it was observed in the intermediate evaluation of those who followed the nutrigenetic diet, for which a low adherence to the protein intake was hypothesized. However, in the final assessment, the participants of the nutrigenetic diet group stabilized their muscle mass; as it were, the muscle mass loss was not sustained. 

The progression from a chronic dysfunction phenotype to a healthy phenotype and vice versa, can be explained by changes in the expression of genetic information or by differences in enzyme and protein activities, which are regulated directly or indirectly by the components of the diet, which entails various progressive processes if the stimulus is maintained until homeostasis or metabolic balance is achieved [35].

At the cellular level, nutrients and food components can act as ligands to activate transcription factors that favor the synthesis of receptors [36]. These molecules can also be metabolized (primary or secondary metabolic pathways), modifying the concentrations of substrates or intermediates or influencing signaling pathways positively or negatively [37].

However, the standard diet based on general recommendations (“diet adapted to all” or “fits all diet”) does not promptly resolve the genetic needs of a population with specific physiological characteristics, which is it has shown a limited effect on reducing the prevalence of CVD and obesity globally [38]. Also, it has been recognized that many people do not achieve lasting benefits in the control of overweight and obesity due to the physiological and neurohormonal adaptation of the body in response to weight loss [39], in addition to experiencing a “weight loss plateau”, and therefore no weight loss occurs [40]. Nonetheless, a longer duration of the intervention would give us a broader picture to confirm this hypothesis or possibly observe compensatory mechanisms in the standard diet group.

Regarding the biochemical parameters, even though the basal values of total cholesterol, LDL-c, glucose, and insulin of the participants were within healthy values, the study population cannot be defined as “metabolically healthy”, according to the criteria exposed by Candi et al. [41], since they presented metabolic risk criteria, such as insulin resistance, (HOMA-IR > 2.5), chronic subclinical inflammation, excess visceral fat, and auto-reported sedentary lifestyles. 

Favorable changes were recorded in most of the biochemical parameters evaluated when comparing baseline values versus final values. However, changes of great clinical significance and greater prominence were detected in the group that followed the nutrigenetic diet. For example, a positive tendency to increase HDL-c was observed in the group that followed the nutrigenetic diet. The fact that the ΔHDL-c resulted in a gain of +1.5 mg/dL in the nutrigenetics group vs. the ΔHDL-c of the standard diet (−0.08 mg/dL) is a highly relevant finding. HDL-c exerts antiatherogenic actions due to direct antioxidant, antithrombotic, anti-inflammatory, and vascular effects, in addition to its essential role in reverse cholesterol transport [42]. Interestingly, it has been shown in models adjusted for covariates that an increase of 1 mg/dL in the HDL-c level reduces the risk of major cardiovascular events by 1.1% [43]. It could be noted that this objective was achieved in this intervention. The increase in HDL-c concentrations of the nutrigenetic diet group is consistent with some studies where a precision diet was administered to individuals with polymorphisms related to hypoalphalipoproteinemia (*CETP*, *ABCA1*, *APOA1*) that followed the nutrigenetic recommendations for each population [17], and an increase in HDL-c of ~1–4 mg/dL is observed in the range of 3–12 months [44,45,46].

Caloric reductions related to rapid weight loss can cause transient reductions in HDL-c levels [47]. But increases in HDL-c, even with the weight loss among the nutrigenetic group, can be explained by various hypotheses: first, individuals with primary hypertriglyceridemia often have reduced HDL-c levels, which is partly due to an exchange between TGs in VLDL and cholesterol esters in HDL; secondly, the result of CETP-mediated activity is a high HDL-TG index (observed in the participants) and a low HDL-c. It should be noted that the nutrigenetic recommendation made by Garcia-Rios et al. [44] was contemplated for the nutrigenetic diet, in which the consumption of a higher % of MUFAs in carriers of the SNP *CETP* rs3764261 promoted a reduction in TGs and an increase in HDL-c. Another hypothesis suggests that the increase in HDL-c may be mediated by increased activity of adipose tissue LPL after weight reductions. In agreement with this hypothesis, Corella et al. [48] recommend the consumption of a diet rich in MUFAs and PUFAs following the Mediterranean diet pattern (MedDiet) for carriers of the SNP *LPL* rs13702 since a decrease in TGs and an increase in HDL-c were observed in the evaluated participants.

The final concentration of HDL-c registered in both groups failed to fall within the reference values marked as healthy (40–60 mg/dL) [49]. This highlights the fact that hypoalphalipoproteinemia [50] is the main dyslipidemia detected in the Mexican population, partly as a consequence of the high prevalence of polymorphisms in the *CETP*, *APOA1*, and *ABCA1* genes [51]. It has been reported that there is an inverse association between body weight and plasma HDL-c [52], and our findings are in agreement; most patients had relatively low HDL-c in their obese state. An alternative hypothesis suggests that the obese state might be associated with a high rate of HDL catabolism; excess adipose tissue accelerates HDL and LDL catabolism [52].

Furthermore, only the nutrigenetic group experienced a significant decrease in TGs (−20% from baseline value) in such a way that the therapy targeted the proposed clinical normal range values according to the guidelines [49]. Clinical studies whose objective is to normalize lipid parameters through a nutrigenetic diet in participants with various polymorphisms related to the metabolism of lipids have described that TG levels decrease ~12–50 mg/dL in a period of 2 to 6 months [33,44].

Obesity is known to increase plasma TG levels, but after starting a calorie-restricted diet TG concentrations generally decrease, mainly due to a reduction in VLDL-TG synthesis, consequent to a low substrate availability and reduced circulating insulin concentrations [52]. An increased clearance promoted by lipoprotein lipase may also be a factor for decreased TGs [53]. Richardson et al. [54] and Corella et al. [48] state that in individuals carrying *LPL* rs13702C, in which lipoprotein lipase activity is decreased, it is necessary to administer poly- and monounsaturated fatty acids (PUFAs and MUFAs) following a MedDiet pattern to reduce TG levels through the proper hydrolysis of TGs, a mechanism modulated by gene–diet interaction.

The triglyceride:HDL-c ratio (TG:HDL) is an easy resource to determine, with a good correlation with the HOMA index in adults, and it has been shown to be an independent predictor of cardiovascular events [27]. The improvement in this index was expected due to the increase in HDL-c and a more significant decrease in TGs registered in the group that followed the nutrigenetic diet compared to the one that followed the standard diet.

The significant improvement in glucose and insulin levels observed in both dietary groups was expected since both treatments were structured with the appropriate amount and type of carbohydrates (CHOs) according to dietary and nutrigenetic recommendations [17,20,55]. The amount and type of CHO determines 90% of the postprandial glycemic response. In addition, studies have shown that controlling carbohydrates in the diet, as well as losing weight, increases insulin sensitivity [56].

The reduction observed in the HOMA-IR implies an increase in insulin sensitivity associated with a decrease in fasting insulin levels, similar to what has been reported in the literature when the caloric intake, refined carbohydrates, and saturated fats are controlled [45,47,57].

It is crucial to mention the effect of adequate MUFAs consumption by the nutrigenetic diet participants, whose primary sources were avocado and olive oil. Recent research attributes beneficial health effects to the oleic acid in avocado and the oleanolic and maslinic acids found in olive oil [58]. In adipose tissue, oleic, oleanolic, and maslinic acids interfere with different steps of adipogenesis and lipolysis by negatively regulating the expression of C/EBPα and PPARγ and by modulating antiadipogenic pathways and kinases [59]. It has been reported that maslinic acid also produced a decrease in cytoplasmic triglyceride droplets, triglycerides, and transcription factors related to the adipogenesis process (PPARγ, aP2) in 3T3-L1 cells [60].

It is well known that obesity is characterized by the elevation of proinflammatory cytokines, including tumor necrosis factor-alpha (TNF-α), interleukin-6 (IL-6), and acute-phase proteins, such as C-reactive protein (CRP), leading to chronic subclinical inflammation [61]. The values of C-reactive protein (CRP) of all participants remained stable throughout the intervention. This finding is consistent with other nutrigenetic interventions or the administration of a Mediterranean diet for up to 9 months [33,57,62]. Gomez-Delgado et al. [63], after one year of administrating the Mediterranean diet to participants with SNP *TNF-α* (rs1800629, rs1799964), observed a significant reduction in this parameter. These results lead to the hypothesis that substantial and sustained diet-induced changes in CRP can occur in the long term.

In the context of chronic subclinical inflammation and adipose tissue dysfunction, the central stimulus for CRP synthesis and release is the action of other cytokines, such as IL-6 and TNF-α [61,64], which is why their measurement is recommended. Scientific evidence shows that adipocyte hypertrophy and hyperplasia are correlated with proinflammatory cytokine concentrations [64,65] and that the reduction in the concentration of the cytokines is consequent to weight loss and the decrease in anthropometric parameters [66]. However, participants who followed the standard diet and lost weight did not have a significant reduction in IL-6 or TNF-α.

The data published in the scientific literature to answer these questions are limited and contradictory. De Luis et al. [67] administered a nutrigenetic diet for three months to carriers of the G308 polymorphism in the *TNF-α*. Still, they did not observe a reduction in the concentrations of proinflammatory cytokines. The aforementioned highlights the importance of using nutrigenetic patterns that consider and encompass various genes involved in a particular metabolic pathway or pathology, as was the case with the panel of 10 genes and 11 genetic variants proposed in this project.

Sureda et al. [68] reported that low adherence to the Mediterranean dietary pattern, rich in MUFAs, fiber, and antioxidants, is directly associated with a worse profile of plasmatic inflammation markers (TNF-α, C-reactive protein, PAI-1, IL-6). Campos Mondragón et al. [69] evaluated the effect of three types of PUFAs (eicosapentaenoic acid EPA+ docosahexaenoic acid (DHA), conjugated linoleic acid (CLA), and walnuts) on subclinical inflammation markers in patients with the metabolic syndrome. They observed that omega-3 (EPA/DHA) supplements reduced the omega-6/omega-3 balance in erythrocytes, suggesting an association with lowering IL-6, leptin, and homocysteine. However, the omega-3 and ellagic acid contained in the walnut could act in synergy since the group that consumed this treatment showed significant reductions in the expression of TNF-α and IL-6 inhibited its proliferation, and, therefore, their concentration in plasma.

In this sense, Ramos-Lopez and Martinez [9] suggest that different types of fatty acids have other effects on the epigenome, giving rise to different physiological responses. Reports indicate that the olive oil intake can regulate the hypomethylation of specific proinflammatory cytokine genes, promoting an anti-inflammatory expression pattern [58]. Voisin et al. [70] studied the effect of the consumption of different proportions of MUFAs, PUFAs, and saturated fats on the DNA methylation of the entire genome, concluding that the amount, ratio, and type of fatty acids is according to the expression of genes that modulate adipogenesis and inflammatory processes. At this point, it is essential to refer to precision nutrition as a reliable tool to determine the requirements and adequate doses of fatty acids based on the genetic needs of individuals.

In addition, the scientific literature indicates that the consumption of fatty acids and certain bioactive compounds can suppress the proinflammatory signaling pathway of Nuclear Factor κ-B (NFκ-B), either by blocking the degradation of the inhibitory protein of NFκ-B in the proteasome or by blocking its phosphorylation, in such a way that the heterodimer is not separated and NFκ-B remains inactivated, does not translocate to the nucleus, and decreases the expression of proinflammatory genes, and, therefore, the synthesis of adhesion molecules and cytokines, such as TFN-α, IL-1β, and IL-6 [66,71].

Conforming to the individual contributions of each predictor into the models, three SNPs showed a direct and significant effect on TG and VLDL levels. Interestingly, these SNPs were specific for standard or nutrigenetic diets, suggesting that the genetic effect on primary outcomes depends on interactions with the macronutrient and fatty acids distribution of the respective dietary patterns. In the standard diet group, LPL rs13702 demonstrated a striking negative effect on the variability in primary outcomes (TGs: β = 159.13 mg/dL, *p* = 0.013; VLDL-c: β = 31.35 mg/dL, *p* = 0.012). The lipoprotein lipase (LPL) breaks down TGs from chylomicrons (CM) and VLDL to be used as energy or stored in fatty tissue for later use [72]. The main factor in the development of hypertriglyceridemia is an impaired clearance, presumably because of LPL dysfunction.

The rs13702 T > C polymorphism is located in exon 10, in the 3’ untranslated region (UTR), position 1672, in the microRNA-410 target site. Goodarzi et al. [73] and Hatefi et al. [74] stated that the existence of the C allele in this position alters the microRNA-410 binding site, and this modification leads to a higher expression of LPL. Therefore, there is an increase in fat metabolites, which accumulate in the liver and muscles and can cause insulin resistance. However, research by Richardson et al. [54] and Corella et al. [48] demonstrated that although the C allele of *LPL* rs13702 disrupts an initial site of the microRNA recognition element for microRNA-410 it results in a gain of function by the enzyme. This SNP was consistently associated with lower circulating TGs and higher HDL-c concentrations. In this framework, the gene–diet interaction that occurs between the SNP rs13702 and the consumption of polyunsaturated fatty acids is highlighted. Since an association with the increase in the expression and activity of LPL is observed, it is proposed that the presence of both factors has a beneficial additive effect, resulting in a decrease in plasma TGs [11,54,74].

*APOA1* rs670 showed a strong positive effect on the ΔTGs and ΔVLDL-c (β = −112.95 mg/dL, *p* = 0.013 and β = −22.39 mg/dL, *p* = 0.012, respectively) of standard diet group participants. There is controversy about the effect of *APOA1* rs670 since, in some studies, the A allele (minor allele) was related to high levels of ApoA1 and HDL-c and normal lipid levels [46]. Still, in other studies no such association was found [75,76]. Scientific evidence shows that rs670 directly affects the response of lipid levels secondary to the amount of fat in the diet [77,78]. Specifically, an enriched polyunsaturated fat diet improved HDL-c and LDL-c and resulted in normal levels of TGs in A-allele carriers [45,79]. In this sense, the quantity and quality of fats (SatFats, MUFAs, PUFAs) could explain the variability in the response to the dietary intervention from interacting with this SNP.

*LIPC* rs1800588 predicted a significative effect on the variability in TGs and VLDL-c when consuming the nutrigenetic diet. The *LIPC* gene codifies the hepatic lipase (HL) enzyme, which is released into the bloodstream to help convert VLDL and IDL into LDL. It also allows HDL to transport cholesterol and TGs from the blood to the liver. In some studies, the CC genotype is considered the risk genotype, while the TT genotype is protective [80]. However, other studies with Mexicans also show that carriers of the TT genotype are strongly associated with hypertriglyceridemia (*p* = 0.0002) and a greater risk of DM2 [81]. Thus, LIPC activity is reported to be determined by sex, body adiposity, dietary fat intake, alcohol consumption, physical activity, and genotype [82].

The cumulative effect of the predictor SNPs, represented in the wGRS, was impressively high. The wGRS predicted TG variability in 69% and VLDL-c fluctuation in 71% following the standard diet. With the consumption of the standard diet, the wGRS predicted 49% and 21% oscillations of TGs and VLDL-c, respectively. Ramos-López et al. [83] observed that GRSs adding risk genotypes are significant predictors of plasma lipid in different performed linear regression models.

There is a clear effect of genetics on the lipid profile, and our findings show that the genetic influence in TG and VLDL-c reduction is related to interactions with diet compositions, so dietary recommendations to control the lipid profile and reduce cardiovascular risk are a fundamental pillar for treating individuals by making modifications in the caloric intake, total fat, MUFAs, PUFAs intake, and contribution of saturated fats.

Reported results on the effect of the GRS on the variance in lipid concentrations (total cholesterol, LDL-c, HDL-c, and TGs) were lower than those obtained in this study. A GRS of lipid-associated SNPs performed by Williams et al. [84] considering individuals of European ancestry accounted for only 3% of the total variance in TGs, while the GRS proposed by Ramos-López [83] predicted the fluctuation in TG levels by 7%. Chang et al. [85] observed the cumulative effect of multiple SNPs (GRS) on the blood lipid levels of non-Hispanic whites, non-Hispanic blacks, and Mexican Americans and reported that the GRS was strongly associated with variability in the lipid profile of the three race/ethnic groups. However, the GRS explained no more than 11% of the total variance in blood lipid levels.

To date, this is apparently the first GRS that includes the panel of 10 genes and 11 SNPs addressed in this study, which were explicitly chosen, taking into account their current high prevalence in the Mexican population, association with the most prevalent dyslipidemias in Mexico, and the robust scientific evidence that supports their effect. Further analyses of interactions (e.g., gene–gene and gene–environment interactions as well as pathway–disease relationships) are definitely necessary to identify other contributors to the blood lipid variance.

Some drawbacks of this research were the sample size and a relatively short follow-up time since recruitment was performed during the COVID-19 pandemic. However, a growing interest among people to improve their lifestyle was observed, even when they experienced the repercussions of confinement on health habits, as has been widely reported [86,87]. The percentage of dropouts in this study agrees with that of other dietary interventions lasting from 4 to 12 weeks (0–35%) [88]. The group that followed the nutrigenetic diet had fewer dropouts than the standard diet group, similar to the numbers reported by Livingstone et al. [38], who observed a greater adherence and commitment on the part of individuals who followed a precision diet vs. a group that was given general dietary recommendations, since they found the precision diet avant-garde and novel. Furthermore, it would be enriching to evaluate the effect that other dietary components, such as fruits and vegetables, as well as lifestyle and environmental factors, undoubtedly provide on biochemical and inflammation markers.

Despite the fact that the results obtained in this research are promising, is important to mention that there are still several challenges and limitations in the field of nutrigenetics and nutrigenomics, both in biomedical research and clinical use. Moreover, translating scientific findings into clinical or public health practices is not immediate. Certain concerns about the clinical practice of nutrigenetics have been reported and represent opportunities for improvement: for example, the proper training of health personnel, which includes nutrigenomic counseling abilities, and the ethical considerations that must be addressed to guarantee the right of access and user protection in direct-to-consumer genetic testing, as well as reducing the costs of genotyping assays and increasing their availability [89,90].

Among the strengths of this study were the use of a nutrigenetic portfolio and the implementation of structured nutrigenetic patterns. To date, this is the first clinical trial that evaluates the effect of a nutrigenetic intervention based on specific nutrigenetic recommendations and that considers a panel of the most prevalent SNPs related to dyslipidemia in Mexicans; also, this study is the first to report the direct effect of specific SNPs and the cumulative effect of a panel of SNPs with a high prevalence in the Mexican population on TG and VLDL-c levels through a GRS.

According to our results, the models presented can help predict the decrease in TGs and VLDL-c after following the standard diet for managing dyslipidemia and weight control, as well as the nutrigenetic diet, according to different nutrigenetic recommendations [17]. In this way, more effective dietary treatment can be prescribed, integrating the genotype and nutrigenetic guidance, especially in individuals with genetic risk. It should be noted that this study included Mexican individuals from western Mexico; for this reason, our findings may not be generalizable to other ethnic groups.

In conclusion, the present nutrigenetic intervention improved lipid abnormalities and inflammatory markers in overweight and obese patients above a functional standard diet. Therefore, given that the variability in the response to the diet has a strong genetic component, precision nutrition offers actionable strategies for reducing the risk of cardiometabolic diseases, representing feasible and adjuvant tools to increase the success of dietary treatments.

## Figures and Tables

**Figure 1 nutrients-15-04324-f001:**
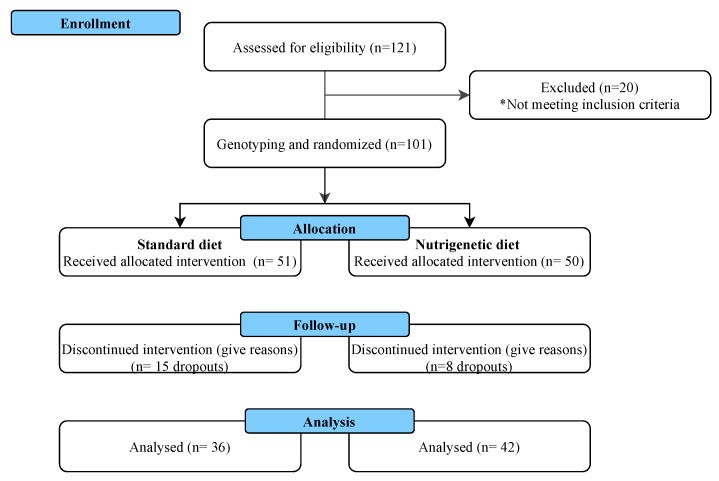
Flow chart describing the follow-up of participants throughout this study.

**Table 4 nutrients-15-04324-t004:** Multiple linear regression models using the panel of SNPs to predict the Δ of TGs and VLDL consumed in the standard diet.

Standard Diet										
	Model 1					Model 2				
	ΔTGs					ΔVLDL				
Predictors	β Coefficients	Std. Error	*p* Value ^a^	PC^2^	*p* Value ^b^	β Coefficients	Std. Error	*p* Value ^b^	PC^2^	*p* Value ^b^
*LIPC* rs1800588	−53.92	30.48	0.127	0.06	0.12	−10.02	5.85	0.138	0.05	0.13
*LPL* rs13702	159.13	45.58	**0.013**	0.26	**0.01**	31.35	8.75	**0.012**	0.26	**0.01**
*FABP2* rs1799883	−28.80	37.05	0.466	0.01	0.46	−5.13	7.11	0.49	0.01	0.49
*CETP* rs708272	43.32	27.69	0.169	0.05	0.16	8.69	5.31	0.153	0.05	0.15
*PPARG* rs1801282	92.52	42.88	0.074	0.10	0.07	17.92	8.23	0.072	0.09	0.07
*APOA1* rs670	−112.95	32.64	**0.013**	0.25	**0.01**	−22.39	6.26	**0.012**	0.26	**0.01**
SATFAT	2.61	1.88	0.216	0.04	0.21	0.51	0.36	0.20	0.04	0.20
MUFAs	−2.33	1.96	0.286	0.02	0.28	−0.44	0.38	0.29	0.02	0.29
Adj. R^2^	0.69		**0.03**			0.71		**0.02**		

Data are expressed as β values ± standard error. Bold numbers indicate *p* < 0.05. PC^2^: squared partial correlation; ΔTGs: delta of triglycerides, ΔVLDL: delta of very low density lipoprotein; *p* value ^a^ = *p* of each predictor; *p* value ^b^ = *p* of PC^2^. Omitted for collinearity: *ABCA1* rs9282541; *APOA5* rs662799; *APOC3* rs5128; and *APOE* rs7412 and rs429358.

## Data Availability

Not applicable.

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
