# Peer review of "A Nutrigenetic Strategy for Reducing Blood Lipids and Low-Grade Inflammation in Adults with Obesity and Overweight"

_nutrients, 2023, doi:10.3390/nu15204324_

Round 1

Reviewer 1 Report

This is an interesting  article where the authors concluded that “This nutrigenetic intervention im- 37 proved lipid abnormalities in patients with excessive body weight. Hence, nutrigenetic strategies could be coadjuvant tools and enhance the standard dietary treatment for cardiometabolic diseases

However, authors must be included the next changes:

Material and method

Please, it is necessary to add  the method to select the sample

Please calculate and include triglyceride glucose index as a marker of insulin resistance

Results

Table 3, please include units of macronutrients

Discussion          

Limitation section must be improved, including problems related with costs of these nutrigenetic test in real-world practice, potential ethical problems with nutrigenetic tests, and so on.

A section of strengths should be included

Author Response

Thank you very much for taking the time to review this manuscript.

Please see the attachment to find the detailed responses and the corresponding corrections highlighted in the resubmitted manuscript.

Reviewer 2 Report

This is an interesting study. however, they present a small number in clinical trial studies. Some parts in the method section need to be improved. 

The comments for the authors are attached below.

Author Response

Thank you very much for taking the time to review this manuscript. Please see the attachment to find the detailed responses and the corresponding corrections highlighted in the re-submitted files.
